# When a Multidisciplinary Approach Is Life-Saving: A Case Report of Cardiogenic Shock Induced by a Large Pheochromocytoma

**DOI:** 10.3390/diseases10020029

**Published:** 2022-05-17

**Authors:** Raffaele Baio, Tommaso Pagano, Giovanni Molisso, Umberto Di Mauro, Olivier Intilla, Francesco Albano, Fulvio Scarpato, Stefania Giacometti, Roberto Sanseverino

**Affiliations:** 1Department of Medicine and Surgery “Scuola Medica Salernitana”, University of Salerno, I-84081 Salerno, Italy; 2Departments of Anesthesiology, Umberto I Hospital, I-84014 Salerno, Italy; tmmpagano@gmail.com (T.P.); francescoalbano@hotmail.it (F.A.); fulvio.scarpato@gmail.com (F.S.); stefaniagiacometti1974@gmail.com (S.G.); 3Department of Urology, Umberto I Hospital, I-84014 Salerno, Italy; molisso.uro@gmail.com (G.M.); umbertodimauro@libero.it (U.D.M.); olivier.intilla@gmail.com (O.I.); roberto.sanseverino@alice.it (R.S.)

**Keywords:** pheochromocytoma, open adrenalectomy, cardiogenic shock

## Abstract

Background: Pheochromocytoma is known for its instantaneous presentation, especially in the younger population. Hemodynamic instability may be the cause of severe complications and impede patients’ ability to undergo surgical treatment. These tumours are surgically difficult to treat due to the risk of catecholamine release during their manipulations, and when they are large, the tumour size represents an additional challenge. In our report, cardiogenic shock developed due to increases in systemic vascular resistance, and the lesion’s size induced surgeons to perform open surgery. Case presentation: A 46-year-old female patient was admitted to our intensive care unit with hypertension and later cardiogenic shock. Systolic dysfunction was noted, along with severely increased systemic vascular resistance. A CT scan showed a left-sided 8.5 cm adrenal mass, which was confirmed as pheochromocytoma using meta-iodobenzylguanidine scintigraphy. Anaesthesiologists and the surgical team planned an effective strategy of treatment. Given the lesion’s size and its apparent invasion of the neighbouring organs, open adrenalectomy (after prolonged hemodynamic stabilisation) was considered safer. The surgery was successful, and the patient remains free from disease two years after the initial event. Conclusions: Large pheochromocytoma can be safely and effectively treated with open surgery by experienced hands but only by seeking to reach hemodynamic stabilisation and minimising the release of catecholamine before and during surgery.

## 1. Introduction

Pheochromocytoma is a rare tumour that arises from chromaffin cells of the adrenal glands and causes excessive production of powerful hormones such as catecholamines. In the past, the most common symptomatology was a classic triad of headache, palpitations, and diaphoresis, as a result of a condition of hypertension (mainly in the form of paroxysmal episodes); however, to date, this triad is hardly ever seen due to the intensive pharmacological treatment of primary hypertension. In effect, only half of these tumours are discovered when analysing blood pressure abnormalities [1]. Occasionally, however, the abundant and prolonged secretion of catecholamines can trigger cardiovascular manifestations (Table 1), causing single or multiple organ failure. This catastrophic condition is termed the “pheochromocytoma crisis” [2]. For this reason, also considering several cases of pheochromocytoma presenting as an acute coronary syndrome or directly as cardiogenic shock, this tumour constitutes a medical emergency in which differential diagnosis is challenging. Furthermore, due to the failure of intensive medical therapy in such cases, emergency adrenalectomy represents the only remaining definitive treatment. In the literature, few reports discuss the emergency surgical treatment of pheochromocytoma. In cases of large and symptomatic adrenal masses, surgery remains the gold standard but which surgical approach (laparoscopy or open surgery) to perform is still a matter of debate [3,4]. First performed by Gagner [5] in 1992, laparoscopic adrenalectomy is reported to have several benefits (in terms of reduction of bleeding, postoperative pain, hospital stay, and convalescence) [6,7,8,9,10,11,12]. Laparoscopic adrenalectomy is indicated in the treatment of several pathologies Table 2 [6]. However, the role of laparoscopy in the treatment of adrenal masses is still controversial when they have a large diameter (5 to 6 cm), are seen invading adjacent organs (e.g., the liver, kidney, or pancreas), or are associated with vein thrombus [7,10,12,13]. However, in centres with high laparoscopic experience, the tumour size is a relative contraindication to laparoscopy [9,14,15]. Owing to radiological imaging’s inability to accurately differentiate malignant from benign lesions, tumour size is regularly used as an indicator of malignant potential; tumours larger than 5 cm are considered at high risk for malignancy [14]. We report the case of a young female patient affected by a large pheochromocytoma which caused “malignant hypertension” and tachycardia, leading to cardiogenic shock. For the patient’s survival, emergency adrenalectomy was performed after a total lack of response to intensive medical treatment. In this report, we discuss the role of open surgery in emergencies and the importance of the multidisciplinary team (MDT) in this context.

## 2. Case Report

A 46-year-old woman suffering from severe headache, nausea, and vomiting was admitted to our hospital emergency department (ED). She had a history of high blood pressure (not under treatment with drugs) and type II diabetes mellitus (under treatment with metformin); she did not report any known drug allergies. However, the anamnestic investigation revealed that the patient had been experiencing episodes of morning vomiting for about one year. For this reason, five months before being admitted to the Emergency Department (ED), the patient had undergone gastroscopy; this, however, did not reveal any abnormalities. On presentation, she was hypertensive (153/87 mmHg) and tachycardic (135 bpm); her respiratory rate was 24 breaths per minute (with respiratory fatigue, which caused generalised cyanosis). We noted that peripheral pulses were barely palpable bilaterally. Later, due to the worsening of the general conditions, the patient was tracheally intubated in order to secure airway patency; thus, the infusion of vasopressor agents was initiated to support circulation. Blood gas analysis showed worsening metabolic acidosis with normal electrolytes. The ECG recorded elevation of the ST segment in the anterolateral regions and poor progression of the R wave in the precordial regions. Blood chemistry test results are shown in Table 3. For these reasons, an emergency coronary angiography was performed. Upon arrival in the haemodynamics room, the patient presented with sinus tachycardia (130 bpm) and a blood pressure reading of 140/70 mmHg. An ultrasound scan was quickly performed while preparing the operative field and revealed the following:-Kinetic alterations (anterolateral akinesia and hypokinesia of the apex);-A moderate depression of the ejection fraction (about 40%);-Ascending aorta of normal size;-Absence of significant valvulopathies;-No evidence of pericardial effusion.

Therefore, diagnostic coronary angiography was performed rapidly, documenting epicardial coronaries free from stenosing lesions. On ventriculography, the cardiac chamber appeared undilated with moderate global contractile dysfunction (FE of approximately 40%), as per Takotsubo syndrome of the mid-ventricular wall. During the procedure, further worsening of haemodynamic conditions (due to a phase of bradycardisation and decreasing pulse oximetry) resulted in pulseless electrical activity (PEA), with a return of spontaneous circulation (ROSC), which was reached after five minutes of cardiopulmonary resuscitation (CPR). Considering the silent medical history of the patient and the abrupt onset of her hypertensive crisis, a total body CT scan was required in order to reach a diagnosis. The brain scan results were negative, while the lungs displayed bilateral pleural effusion (maximum thickness 11.5 mm on the right side and 10.7 mm on the left side), with associated atelectasis of the left lung parenchyma. Finally, we found an 8.5 cm adrenal mass which occupied the superior cap of the left kidney. This mass displaced the pancreas tail up and forward, compressing the left kidney’s upper pole. In addition, there was a blood-type flap in the left perirenal area, as seen from the recent bleeding shown in Figure 1. The [123I]-MIBG scintigraphy was also performed, which showed intense emission from a left-sided adrenal mass. The patient was admitted to intensive care because of the persistent haemodynamic instability, despite volume filling with crystalloids and norepinephrine infusion. The echocardiogram performed in intensive care showed a further reduction in the ejection fraction to 25%. As a further complication caused by refractory haemodynamic instability, acute renal failure was superimposed (treated by continuous renal replacement therapy). Continuous infusion of levosimendan was added as per protocol for the treatment of cardiogenic shock. Blood chemistry results are shown in Table 4. The dosage of catecholamines, vanillylmandelic, and homovanillic acid in the urine was not possible because it took 5–7 days. Considering the clinical situation and laboratory results (which showed increased blood levels of metanephrine and normetanephrine), which strongly indicated pheochromocytoma, an alpha-blocking therapy (Phenoxybenzamine, from 10 up to 80 mg/die) was initiated. After four days, despite the pharmacological treatments, haemodynamic instability persisted, with phases of marked hypotension alternating with phases of refractory hypertension. For this reason, an MDT was assembled to manage the patient. Urologists, radiologists, and anaesthesiologists agreed on a course of action: an open surgery performed under general anaesthesia. Monitoring included radial and pulmonary artery catheters; as expected, during tumour isolation, the patient developed a hypertensive crisis which was managed with continuous infusion of sodium nitroprusside and esmolol. We chose an anterior transperitoneal approach through bilateral subcostal laparotomy according to Chevron, which provided the surgeons with optimal exposure and confirmed this surgical approach’s usefulness in the excision of larger adrenal masses. Left kidney compression emerged during a direct inspection. After careful separation of surrounding structures and identification of the left renal vein, the adrenal vein was isolated and ligated in order to block the incretion of catecholamines, followed by the artery. The tumour’s size and the inflammatory reaction complicated cleavage, but the removal of the large mass was completed in 30 min, with an estimated blood loss of 150 mL. Hydrocortisone was administered to manage the expected transient acute adrenal insufficiency in the first 48 h. Hypotension developed early when the renal vein was clamped, and its management required norepinephrine infusion, in addition to discontinuation of the previous short-acting vasoactive drugs. Norepinephrine was gradually discontinued within the first 18 h. Hydrocortisone was tapered to 100 mg/day (from 200 mg daily) and replaced with same-dose oral cortisone; total cortisol and ACTH levels were monitored to guide therapy and were maintained within normal ranges. Histopathology revealed an adrenal pheochromocytoma characterised by solid and alveolar growth, moderate nuclear pleomorphism with the presence of hyaline blood cells, and rare mitosis (immunohistochemical examination: CK-, VIM-, EMA-, CD56+, synaptophysin+, chromogranin+, neuron-specific enolase (NSE)+, Ki67 equal to 1%). The neoplasm appeared delimited by a fibrous pseudo-capsule and showed almost complete haemorrhagic infarction with blood extravasations in the surrounding adipose tissue. There were no images of angioinvasion or adrenal gland infiltration (PASS score: 3). Without any metastases or recurrent symptoms or signs, the MDT maintained a close follow-up and withheld further treatment. The intensive postoperative hospitalisation continued until the stabilisation of the patient’s clinical condition and her transfer to the hospital ward. The patient also tested negative on genetic tests for familial syndromes that could cause the development of pheochromocytoma. Two years after diagnosis and surgery, she enjoys a disease-free life. The close follow-up includes urine catecholamine tests twice a year and magnetic resonance every two years. Metabolic tests were normalised since the first check-up visit, 6 months after surgery.

## 3. Discussion and Conclusions

Pheochromocytomas are rare neuroendocrine tumours, occurring in 0.1–2% of people with hypertension, while the incidence rises to 4–5% in patients with incidental adrenal mass [16,17]. Furthermore, autopsy studies report a relatively high prevalence of these neoplasms. This suggests that many tumours go undiagnosed, resulting in sudden death or premature mortality [18]. Pheochromocytoma occurrence may be sporadic [19] but more typically arises in patients with inheritable syndromes. The most common forms of pheochromocytoma are sporadic (90%), usually affecting patients between the ages of 40 and 50. However, in association with familial syndromes, hereditary forms are also to be considered (Table 5); these forms usually affect patients before age 40. MEN 2 (2A and 2B) is an autosomal dominant disease which predisposes to the development of several tumours derived from the neural crest tissue [20]. In MEN 2A pheochromocytoma is associated with MTC (Medullary Thyroid Cancer) and the primary hyperparathyroidism (PHPT) (caused by adenoma and/or parathyroid hyperplasia), while there is no presence of PHPT in MEN 2B. MEN 2A and 2B are caused by a RET mutation, which codifies for a tyrosine kinase transmembrane receptor; its ligand is the neurotrophic factor. VHL disease is recognised as a hereditary cancer syndrome predisposing to the development of highly vascularised multiple endocrine tumours. More than 20–25% of these patients develop pheochromocytoma, frequently multifocal and bilateral, with a risk of malignancy super-imposable to those of MEN 2A [21]. The VHL codifies for two proteins (pVHL 19 and pVHL 30) and the loss of function of pVHL results in angiogenic activation (which explains the high vascularisation of VHL tumours). In VHL patients, pheochromocytoma can often remain “silent” or can present few symptoms. Furthermore, in these patients, the age of onset of pheochromocytoma can be earlier with respect to sporadic pheochromocytoma and can manifest itself with multiple lesions. NF1 is a relatively frequent genetic autosomal dominant disease. The NF1gene is localised on the long arm of chromosome 17, and its product is the neurofibromin. It is expressed both in the cells deriving from the neural crest and in other tissues with a different embryonal derivation. This protein acts as an onco-suppressor, inhibiting the activation of the proto-oncogene RAS, and thus regulating cellular proliferation and differentiation. In these patients, pheochromocytoma has an incidence that is 10-fold higher than the general population, while paragangliomas are less common. Furthermore, patients with NF1 present bilateral pheochromocytoma more frequently than the general population, and the malignant forms occur in 11% of cases [22]. The so-called paraganglioma of the neck/pheochromocytoma syndromes, or PGL1, is caused by a germline mutation of the gene encoding the D subunit of SDH, mitochondrial complex II, involved in the Krebs cycle and in the aerobic electron transport chain [23]. The gene for the SDHD, located in chromosome 11, codifies for an anchoring protein [24]. These patients are characterised by the presence of multiple paragangliomas of the head and neck, associated with intra- and extra-adrenal pheochromocytoma. PGL 4 syndrome is due to germline mutations of the SDHB gene located on chromosome 1. SDHB mutations have been strongly associated with extra-adrenal pheochromocytoma, particularly in the abdomen and with malignancy. Head-and-neck PGLs have been observed less frequently [25].

In the majority of cases, pheochromocytoma is clinically silent, presenting as adrenal “incidentaloma” [26]. However, in patients affected by this tumour, the differential diagnosis is very complex due to the wide range of clinical symptoms reported [27]. For this reason, pheochromocytoma is considered the great mimic. In effect, 77–98% of patients with pheochromocytoma suffer from hypertension.

This tumour may constitute a medical emergency, mainly in case of complications [28]. Some patients with a background of hypertension show unexplained orthostatic hypotension; similar conditions are helpful in making a correct diagnosis. A shock after hypotension is highly probable and usually due to several molecular events (Table 6). (1) Volume depletion, or contraction of extracellular fluid volume, occurs as a consequence of total body sodium loss. As water crosses plasma membranes in the body through passive osmosis, the loss of the major extracellular cation (Na) rapidly also results in water loss from the extracellular fluid space. For this reason, sodium loss always causes water loss. Common causes of volume depletion are vomiting, sweating, diarrhoea, burns, use of diuretics, and kidney failure. When fluid loss is less than 5% of the extracellular fluid volume (mild volume depletion), the only sign may be decreased skin turgor (best assessed at the upper torso). The patient may feel thirsty. Dry mucous membranes do not always correlate with volume depletion, especially in elderly patients or those who breathe through the mouth. Oliguria is typical. When the volume of extracellular fluid has decreased by 5–10% (moderate volume depletion), orthostatic tachycardia, hypotension, or both are generally, but not always, present. Skin turgor may further decrease. When fluid loss is greater than 10% of extracellular fluid volume (severe volume depletion), signs of shock may occur (e.g., tachypnoea, tachycardia, hypotension, confusion, and poor capillary filling). (2) Tissues made up of tumour cells need continuous vascularisation for their growth. A tumour goes into necrosis when it grows excessively and the insufficient presence of blood vessels causes its necrosis. In the case of pheochromocytoma, which is a secreting tumour, the loss of tumour tissue causes abrupt cessation of the release of catecholamines (with the consequent risk of the onset of hypotension and shock). (3) Like channel receptors, G protein-coupled receptors (adrenergic receptors belong to this family of receptors) also undergo desensitisation, even if it is different desensitisation. In fact, while the receptor channel desensitises rapidly, as this is an intrinsic property of the receptor itself and is easily inactivated following the action of the ligand, in the case of G proteins, there is a loss of the receptor response due only to the chronic action of G protein-coupled receptor agonists. In the case of pheochromocytoma, the continuous and prolonged secretion of catecholamines over time can lead to desensitisation of the receptors, with the consequent cessation of adrenergic effects and the risk of hypotension or even shock. (4) In the context of pheochromocytoma, it is also possible to demonstrate the presence of cells that produce neuropeptides with immunohistochemical methods [29]. Among these, met-enkephalin (already present in the normal medulla) and calcitonin (which can also be secreted) are found with some frequency [30]. Calcitonin increases the renal excretion of phosphorus and stimulates the reabsorption of calcium, favouring its deposition in the bones. This inevitably results in a reduced plasma concentration of calcium, a condition known as “hypocalcaemia”. The cardiac manifestations of hypocalcaemia include hypotension, myocardial contractility deficit with reduced cardiac output, bradycardia, arrhythmias, decreased sensitivity to digitalis, refractory heart failure, and increased incidence of cardiovascular events (mainly ischemic heart disease). The secreted catecholamines’ strong effect is considered the cause of potentially lethal cardiovascular complications in patients with these tumours [31]. In these patients, despite coronary–artery normality, acute heart failure with pulmonary oedema is possible [32]. The presence of pheochromocytomas may cause the same electrocardiographic changes as acute myocardial infarction [33,34], malignant cardiac arrhythmia, and even dissecting aortic aneurysm. Sudden death, heart failure (due to toxic cardiomyopathy), hypertensive encephalopathy, acute cerebrovascular event, or neurogenic pulmonary oedema are other possible cardiovascular complications in patients with pheochromocytoma [35,36,37].

Assessing the excessive production of catecholamines and/or their metabolites is crucial for the diagnosis of pheochromocytoma. Measurements can be carried out both in blood and urine; the most commonly measured compounds are the catecholamines themselves (noradrenaline and adrenaline), the metanephrines (normetanephrine and metanephrine), and the vanillylmandelic acid (VMA). Recent reports indicate the measurement of plasma or urinary fractionated metanephrines is the most sensitive tool for the diagnosis of pheochromocytoma [38,39,40]. In effect, metanephrines have a longer half-life than adrenaline and noradrenaline, whose half-life is a few minutes. The sensitivity of the measurement is to be considered the most important information in this diagnostic process where, due to the danger of the tumour, false-negative results should be avoided. The measurement of metanephrines involves a certain number of false positives but, given the same specificity as that of the other metabolites, it remains the most sensitive measurement [40]. In effect, considering that catecholamine release is often paroxysmal, a single measurement may be inadequately sensitive. Repeating tests two or more times helps improve the sensitivity, especially following a paroxysmal episode. Therefore, the advantage of measuring metanephrines lies in the fact that these are always secreted by the tumour in a continuous way, with a different and autonomous mechanism, compared with that of catecholamines [41] which, on the other hand, are often of the intermittent type. For this reason, the determination of normal values of metanephrine and normetanephrine in the plasma or urine allows physicians to exclude the presence of pheochromocytomas in practice. Doubtful increases in metanephrine concentrations require further investigations, ranging from the repetition of baseline measurements to the use of dynamic tests. Among these, the glucagon stimulus test [42] must be considered obsolete due to its low sensitivity and specificity and, above all, its dangerousness. Conversely, the clonidine suppression test [43] is a non-dangerous test, which may be useful to better define diagnosis in patients with borderline increases in catecholamines or metanephrines. The test is based on the principle that chronic activation of the sympathetic system, which involves an increase in the release of catecholamines and an increase in metanephrines for their peripheral metabolisation, is affected, unlike the tumour secretion, by the inhibitory action exerted by clonidine through the stimulation of presynaptic alpha-2 receptors. It follows that increased levels of catecholamines caused by sympathetic–adrenergic activation are brought back to normal values by clonidine which, on the other hand, has no effect on increased levels due to tumour secretion. The test involves the measurement of plasma catecholamines or metanephrines, is easily practicable, and has few contraindications. Among these, its execution in patients treated with beta-blockers in whom clonidine could induce excessive bradycardia or brachyarrhythmias. Furthermore, it should be borne in mind that the test cannot be performed in patients treated with tricyclic antidepressants, in which the values of catecholamines and metanephrines are often higher than normal. In fact, tricyclic antidepressants, through inhibition of neuronal reuptake, increase the inter-synaptic levels of noradrenaline and cause downregulation of presynaptic alpha-2 receptors, thus reducing the action of clonidine. In such patients, the test gives false-positive responses. The withdrawal period of antidepressants necessary to re-establish sensitivity to clonidine is relatively long and is around three to four weeks. In conclusion, for the diagnosis of pheochromocytoma, the measures to be preferred are those of plasma or urinary fractionated metanephrines. Plasma catecholamines offer good indications, especially if the results are interpreted in the light of the patient’s clinical condition at the time of sampling or urinary collection. The dynamic tests are to be reserved for very selected cases and are limited to that of inhibition with clonidine.

Therefore, the patient’s blood chemistry results (Table 4) when she was admitted to intensive care showed values of 7.260 and 7.860 pg/mL for metanephrine and normetanephrine, respectively, which are highly diagnostic of pheochromocytoma. Once the laboratory investigations have shown alterations of catecholamines and their metabolites, it is possible to use different imaging techniques. Nuclear magnetic resonance (which identifies 90% of adrenal masses) and scintigraphy with meta-iodo-benzyl-guanidine (owing to the high affinity of this tracer for chromaffin tissues) represent valuable tools for the diagnosis. An abdominal CT scan is crucial in emergencies; when it identifies an adrenal mass in cases of strong clinical suspicion, the diagnosis of pheochromocytoma is highly probable.

Regarding surgical treatment, elective surgery (associated with an appropriate preoperative medical therapy) is the best option, resulting in less than 1% operative mortality when an experienced anaesthesiologist and a skilled surgeon work together [44]. However, in cases of shock which prevent haemodynamic stabilisation with medical treatment, emergency surgery is the only option to halt progressive multiple organ failure. Hypertensive crises, cardiac arrhythmias, pulmonary oedema, and cardiac ischemia are catecholamine-induced and potentially lethal complications during surgery; the scope of medical pretreatment is to prevent these complications with the blockade of alpha-adrenoceptors. Blocking alpha-adrenoceptors non-competitively, phenoxybenzamine is the most preferred drug for this purpose. In most cases of pheochromocytoma, hypertension is the typical symptom; however, in some patients lacking a trigger event, its unexpected onset may be mistaken for more common diseases for that population. In our patient, acute coronary syndrome provided the first diagnostic suspicion; the successful management of her hypertensive crisis and subsequent shock was the result of teamwork between different specialists. In the management of patients with pheochromocytoma, the choice of surgical approach is another crucial consideration. First described by Gagner et al. [5] and gradually becoming the gold standard in cases of large adrenal masses [45,46,47], now laparoscopic adrenalectomy is the preferred surgical technique at experienced centres. In effect, compared with open surgery, the laparoscopic approach reduces postoperative morbidity, hospital stay, and expense [6,48,49], with a low rate of both complications and conversion [50]. However, open surgery is mandatory in emergencies (i.e., when, in cases of haemodynamic instability, rapid action is crucial to patient survival). In choosing the surgical approach, the size of the pheochromocytoma is another important aspect to consider [51]. For “large” tumours, such as pheochromocytoma, with a diameter greater than 6 cm, laparoscopic adrenalectomy is technically difficult to perform but feasible [52,53,54]. Conzo et al. [55], as well as de Fourmestraux and Coll [52], retrospectively reviewed the results of laparoscopic adrenalectomy in large and normal (diameter < 6 cm) adrenal masses; no significant differences in terms of blood loss or surgical time were reported. According to more current case series, laparoscopic surgery engenders significantly extended surgery time and greater blood loss [3,4]. Furthermore, in patients with this adrenal mass, intraoperative hypertension or tachycardia is often the result of tumour manipulations, and laparoscopy probably does not bring greater haemodynamic stability [56,57]. This may be surprising if we consider that, compared with open surgery, the laparoscopic approach is “gentler” on the mass. However, hypotension should occur in both open and laparoscopic surgery; this is because it is initially associated with the interruption of catecholamine incretion (due to adrenal vein ligature) and, later, with relative adrenal insufficiency. Our patient was affected by a tumour 8.5 cm in diameter—very large according to common criteria—which compressed the upper pole of the left kidney and the tail of the pancreas. Although the surgeon (R.S.) had an excellent record with laparoscopic adrenalectomy, the open surgery allowed us to minimise tumour mobilisation until the adrenal vein was clamped (this was chosen considering that, in cases of such a large mass, laparoscopy would not have guaranteed reduced manipulation of the tumour). Our patient was affected, despite the absence of a trigger event, by cardiogenic shock due to vasoconstriction. For this reason, during surgery, the surgeon’s main concern was to minimise catecholamine discharge. Regarding surgery time, most authors advice emergency adrenalectomy in cases of progressive worsening of the patient’s haemodynamic status or in cases of multiple organ failure that fail to respond to maximal medical treatment [58,59]. After surgery, considering the two major postoperative complications (hypotension and hypoglycaemia), close surveillance of the patient for the first 24 h is mandatory. Considering the continuous blockade of alpha-adrenergic receptors (by phenoxybenzamine), postoperative hypotension is associated with the abrupt drop in circulating catecholamines after tumour removal. In this case, management consists of fluid replacement and occasionally intravenous administration of ephedrine. The onset of hypoglycaemia is associated with rebound hyperinsulinemia due to the recovery of insulin release after tumour removal. Few reports show reversal of cardiomyopathy and its symptoms after tumour removal [31,60]; however, if the pheochromocytoma remains occult over a long period, a heart transplant is the only definitive treatment.

In conclusion, the catastrophic presentation of our patient’s pheochromocytoma was unusual because her cardiogenic shock did not show the typical aspects of adrenergic cardiomyopathy (also known as Takotsubo syndrome); in patients with pheochromocytoma, this type of cardiomyopathy is typically focal [61]. Analysing 80 case reports of pheochromocytoma-associated Takotsubo syndrome, globally reduced systolic function was present in 20% of cases [62]; however, out of 1750 patients in the International Takotsubo Registry, none had global systolic depression (while in all patients, focal ballooning was reported) [63]. As revealed by our study, and as demonstrated by other authors in similar cases [62], the acute heart failure in our patient was the result of severely increased systemic vascular resistance. In conclusion, we report a case of pheochromocytoma which rapidly evolved into cardiogenic shock secondary to increased afterload. An MDT selected open adrenalectomy as the safest surgical approach after considering the large size of the adrenal mass and its secretory potential. Our experience is unique for the clinical presentation of the case but also because the simultaneous involvement of several specialists (e.g., the anaesthesia team for preoperative and intraoperative hemodynamic balance, the surgical team) was crucial for successful management. In effect, our case shows that a multidisciplinary approach is life-saving in the treatment of decompensated pheochromocytoma which manifests itself with severe cardiological complications.

## Figures and Tables

**Figure 1 diseases-10-00029-f001:**
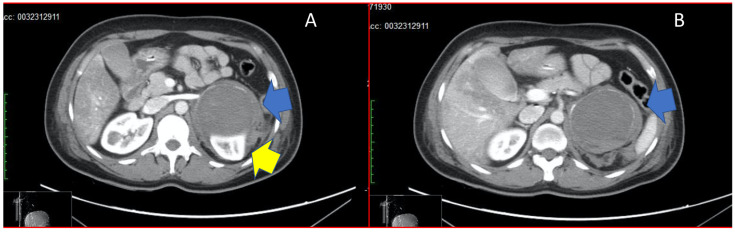
Panel (**A**) shows adrenal mass (blue arrow) compressing the upper pole of the left kidney (yellow arrow). Panel (**B**) shows the adrenal mass in its total size, with haemorrhagic infarction (blue arrow).

**Table 1 diseases-10-00029-t001:** Cardiovascular manifestations associated with pheochromocytoma.

Malignant Arrhythmia
Cardiomyopathy
Acute coronary syndrome
Acute heart failure

**Table 2 diseases-10-00029-t002:** Indications for adrenalectomy.

Pathological Conditions
Aldosteronoma
Pheochromocytoma
Cortisol producing adenoma
Nonfunctioning adenomas
Rare entities (cysts and myelolipomas)

**Table 3 diseases-10-00029-t003:** Blood chemistry test results when the patient was admitted to Emergency Department.

	Patient Value	Normal Range
**White blood cell count**	15.400	4.000–10.000/mm^3^
**Troponin I**	64.7	≤40 ng/L
**Myoglobin**	162.7	14.3–65.8 ng/mL
**Blood glucose**	360	60–100 mg/dL
**Lactate**	8.9	<4 mEq/L
**CK-MB**	3.10	0.6–6.3 ng/mL

**Table 4 diseases-10-00029-t004:** Blood chemistry results when the patient was admitted to intensive care.

	Patient Value	Normal Range
**Cortisolemia**	65	6.2–19.4 mcg/dL (in the morning) 2.3–11.9 mcg/dL (in the afternoon)
**Adrenaline**	340	20–190 pg/mL
**Metanephrine**	7.260	0–90 pg/mL
**Normetanephrine**	7.860	0–180 pg/mL
**Renin**	310	3–40 pg/mL

**Table 5 diseases-10-00029-t005:** Familial syndromes with hereditary forms of pheochromocytoma.

Multiple endocrine neoplasia type II
Von Hippel–Lindau syndrome
Neurofibromatosis type I
Carney syndrome
PGL 1 syndrome
PGL 4 syndrome

**Table 6 diseases-10-00029-t006:** Molecular events associated with shock after hypotension.

Intravascular volume depletion
Abrupt cessation of catecholamine secretion due to tumour necrosis
Desensitisation of adrenergic receptors
Hypocalcaemia

## Data Availability

The study did not report additional data to those described in the text. All data generated or analysed during this study are included in this published article.

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
