# Peer review of "When a Multidisciplinary Approach Is Life-Saving: A Case Report of Cardiogenic Shock Induced by a Large Pheochromocytoma"

_diseases, 2022, doi:10.3390/diseases10020029_

Round 1
Reviewer 1 Report
Pheocrhormocytoma is a tumor originating from adrenal medulla chromaffin cells that synthesize, store, and catabolize catecholamines. The typical triple symptoms of pheochromocytoma include recurrent headaches, massive sweating, and palpitations. Hypertension is also one of the most common manifestations in patients with pheochromoctomas. In this work, the Authors reported a very interesting case report. Moreover, the Authors well described the limits of the study in the final section. The Authors should just add some brand new quotes regarding the prevalence, and also new quotes about family syndromes with hereditary form of pheochromocytoma. The Authors ahouls also specify the antihypertensive drugs taken by the patients before the surgery. Moreover, has the patient been prepared according to the guideline for the adrenalectomy? In the end, we sugget to read the study “Clinical experience with pheochromocytoma in a single centre over 16 years” (doi: 10.2165/11530430-000000000-00000. Epub 2013 Jan 3.) in order to compare the informations deduced by the two studies. The study is well conducted and structured, the results and conclusions are very interesting and very important.
Author Response
We thank the reviewer for appreciating our manuscript. As requested, we have modified the text, including information on the prevalence of pheochromocytoma and associated hereditary familial syndromes. Furthermore, we specified that the patient was not taking any type of antihypertensive drug. The patient was prepared for surgery according to the guidelines.
Reviewer 2 Report
The technical note by Baio et al. presented a case of a 46-year-old female patient who was admitted to their intensive care unit with hypertension and later cardiogenic shock. A CT scan showed an adrenal mass which was confirmed as pheochromocytoma. These tumors are surgically difficult to treat due to the risk of catecholamine release during their manipulations but given the extended lesion’s size and its apparent invasion of the neighbouring organs, an open adrenalectomy was considered safer. Surgery went well and the patient remains free from disease two years from the initial event. The authors concluded that large pheochromocytoma can be effectively treated with open surgery by experienced hands, keeping attention to reach hemodynamic stabilization and minimizing the release of catecholamine before and during surgery.
The article precisely described the clinical case and could be very interesting from a multidisciplinary point of view.
Only few minor concerns should be addressed by the Authors, before a publication on Diseases can be granted:
- The “Discussion and Conclusions” paragraph is too long and difficult to follow, thus the Authors need to divide it in subparagraphs: first the description of pheochromocytoma, then the risks associated to this type of tumor, next the tests used for the diagnosis and in the end the surgical approach and their conclusion;
- The Authors’ names must be written in uppercase at the beginning of the article;
- Check typing and grammar mistakes and use the same font throughout the text.
Overall. MINOR REVISIONS are required.
Author Response
We thank the reviewer for appreciating our manuscript. As requested, we have divided the “discussion and conclusions” paragraph into five sub-paragraphs, following the instructions of the reviewer. Also, as required, the authors' names have been capitalized and the same character has been applied throughout the text.
This manuscript is a resubmission of an earlier submission. The following is a list of the peer review reports and author responses from that submission.
Round 1
Reviewer 1 Report
The manuscript submitted is an interesting case report of a pheochromocytoma presenting with cardiogenic shock.
The strengths of this work are the unicity of the case reported and the multidisciplinary approach used. I appreciated the article, yet some minor revisions need to be addressed:
- In the abstract, it is reported that the diagnosis of pheochromocytoma was confirmed using meta-iodobenzyl-19 guanidine scintigraphy, yet this information is not present in the main text. Please revise it.
- Also, in the abstract, it is stated that “the patient remains free from disease two years from the initial event”, but I could not find this information in the case presentation, nor any data about patient’s follow up were given. Please add them.
- In the text, it is not completely clear how the diagnosis of pheochromocytoma was achieved. In my opinion, the biochemical results in Table 4 should be explained and discussed, taking also into account the possible interfering medications and clinical conditions.
- Analogously, the events reported in Table 6 should be described in the text.
- The abstract focuses on how haemodynamic stabilization guaranteed a good prognosis for the patient. For that reason, in the case presentation, it should be explained more precisely how haemodynamic stability was achieved and which pharmacological interventions were used.
- I would suggest improving English style: lexicon is somehow poor and some spell mistakes require your attention.
Author Response
"Please see the attachment"

Reviewer 2 Report
The authors have presented a very interesting case report of an urgently operated pheochromocytoma. However, there are some major issues that need to be addressed.
Abstract: Line 14: the pheochromocytomas are risky to operate due to catecholamine excess not only when they are large in size. Please be more precise in order to avoid misenterpetations.
Introduction: Line 30: please remove the word mostly.
Line32: the classic triad of pheochromocytoma includes headache, palpitations and diaphoresis
Line 58: please clarify the term 'catastrophic hypertension'. Maybe the term 'malignant hypertension' would be more suitable.
Case: Line 113: please report the dosage and the type of a-blocker used.
Line 128: Please clarify the reason of hydrocortisone administration. Did you have any signs of ACTH secretion from the pheochromocytoma? The right adrenal was normal?
Discussion: Line 177-180: the gold standard for the biochemical diagnosis of pheochromocytoma is the measurement of plasma and/or urine metanephrines. The VMA levels are not sensitive nor specific.
Table 2: The table reports the indications of adrenalectomy and not specifically of laparoscopic adrenalectomy. It would be preferable to replace the term 'Cushing's disease' with the term 'cortisol producing adenoma'. There is no indication to operate non-functioning adenomas unless there is suspicion of malignancy.
Table 5: there are additional familial syndromes that are related to pheochromocytomas and should be reported.
Author Response
"Please see the attachment"

Round 2
Reviewer 2 Report
I would like to thank the authors for their effort to provide a revised version of the manuscript. However, apart from the linguistic issues there are still some flaws and points that need improvement. In my point of view, the paper can not be accepted in the present form.